# Vitamin D: Beyond Traditional Roles—Insights into Its Biochemical Pathways and Physiological Impacts

**DOI:** 10.3390/nu17050803

**Published:** 2025-02-26

**Authors:** Vlad Mihai Voiculescu, Andreea Nelson Twakor, Nicole Jerpelea, Anca Pantea Stoian

**Affiliations:** 1Department of Dermatology, Carol Davila University of Medicine and Pharmacy, 050474 Bucharest, Romania; vlad.voiculescu@umfcd.ro (V.M.V.); nicole.jerpelea@stud.umfcd.ro (N.J.); 2Department of Dermatology, Elias University Emergency Hospital, 011461 Bucharest, Romania; 3Internal Medicine Department, “Sf. Apostol Andrei” Emergency County Hospital, 145 Tomis Blvd., 900591 Constanta, Romania; 4Department of Diabetes, Nutrition and Metabolic Diseases, Carol Davila University of Medicine and Pharmacy, 050474 Bucharest, Romania; anca.stoian@umfcd.ro

**Keywords:** vitamin D, skin, dermatology, deficiency, supplementation, sunshine vitamin

## Abstract

**Background:** It is true that vitamin D did not earn its title as the “sunshine vitamin” for nothing. In recent years, however, there has been a shift in the perception surrounding vitamin D to a type of hormone that boasts countless bioactivities and health advantages. Historically, vitamin D has been known to take care of skeletal integrity and the calcium–phosphorus balance in the body, but new scientific research displays a much larger spectrum of actions handled by this vitamin. **Materials and Methods:** A systematic literature search was performed using the following electronic databases: PubMed, Scopus, Web of Science, Embase, and Cochrane Library. **Results:** Many emerging new ideas, especially concerning alternative hormonal pathways and vitamin D analogs, are uniformly challenging the classic “one hormone–one receptor” hypothesis. To add more context to this, the vitamin D receptor (VDR) was previously assumed to be the only means through which the biologically active steroid 1,25-dihydroxyvitamin D3 could impact the body. Two other molecules apart from the active hormonal form of 1,25(OH)2D3 have gained interest in recent years, and these have reinvigorated research on D3 metabolism. These metabolites can interact with several other nuclear receptors (like related orphan receptor alpha—RORα, related orphan receptor gamma—RORγ, and aryl hydrocarbon receptor—AhR) and trigger various biological responses. **Conclusions:** This paper thus makes a case for placing vitamin D at the forefront of new holistic and dermatological health research by investigating the potential synergies between the canonical and noncanonical vitamin D pathways. This means that there are now plentiful new opportunities for manipulating and understanding the full spectrum of vitamin D actions, far beyond those related to minerals.

## 1. Introduction

The oldest hormone on Earth, vitamin D, is linked to the health of practically all living things, from phytoplankton to people [1]. It is most usually associated with musculoskeletal system maintenance; however, its biological features go far beyond calcium–phosphorus equilibrium [2]. Even so, this fat-soluble vitamin with steroid hormone properties is pivotal in regulating calcium and phosphate homeostasis, which are essential for skeletal health. Its influence also include roles in immune modulation, cellular differentiation, and the regulation of metabolic pathways [3]. Vitamin D exists in two primary forms: D2 (ergocalciferol), derived from plant sterols, and D3 (cholecalciferol), synthesized in the skin due to UVB light exposure or obtained from animal-based dietary sources [4].

Figure 1 shows that the skin plays a critical role in vitamin D synthesis. UVB radiation (wavelength 290–315 nm) converts 7-dehydrocholesterol in the epidermis into pre-vitamin D3 [5]. It is a compound of thermal isomerization for transformation into cholecalciferol. The vitamin D-binding protein (DBP) carries it to the liver, where cholecalciferol is hydroxylated to 25-hydroxyvitamin D [25(OH)D] [6,7]. It is the main circulating metabolite. Further hydroxylated in the kidney by a second hydroxylase to biologically active 1,25−dihydroxyvitamin D [1,25(OH) 2D], this metabolite binds to the VDR [8,9]. Figure 1 below shows the Vitamin D metabolism and systemic regulation pathway

In addition to regulating skeletal activity, vitamin D ameliorates cardiovascular functioning. It plays a role in preventing hypertension caused by the regulation of the renin-angiotensin system and slowing down autoimmunity and certain cancers through cell cycle and apoptosis control [6,11,12].

Through the vitamin D receptors in the brain, vitamin D can protect neurons in the central nervous system whilst improving cognitive capacity and reducing the incidence of neurodegenerative diseases [9]. All these problems lead to the results of vitamin D deficiency becoming epidemic; an increasing reliance on caffeinated products, low exposure to the sun, dietary restriction, and even the most regular physical activity and the use of sunscreen all factor into this [13].

Deficiency is associated with conditions like rickets, osteomalacia, cardiovascular diseases, autoimmune disorders, and increased susceptibility to infections [2,11].

## 2. Materials and Methods

### 2.1. Literature Search and Study Selection

This systematic review was conducted following the PRISMA (Preferred Reporting Items for Systematic Reviews and Meta-Analyses) guidelines to ensure transparency and reproducibility in the literature search and selection process.

A systematic literature search was performed using the following electronic databases: PubMed, Scopus, Web of Science, Embase, and Cochrane Library. The search strategy was constructed using a combination of MeSH terms, keywords, and Boolean operators (AND, OR, NOT). The primary search terms included “Vitamin D metabolism”, “Lumisterol”, “Tachysterol”, “Vitamin D pathways”, “Skin metabolism”, and “Non-canonical vitamin D synthesis”. The search was limited to peer-reviewed articles published in English from 2000 to 2025.

### 2.2. Eligibility Criteria

Inclusion Criteria

Studies were included if they met the following criteria:-Investigated vitamin D metabolism, lumisterol, or tachysterol;-Explored biochemical pathways and physiological effects;-Conducted in humans, animal models, or in vitro systems;-Published in peer-reviewed journals.

Exclusion Criteria

The exclusion criteria included the following:-Publications not written in English;-Studies focusing solely on dietary intake or supplementation without biochemical pathway analysis;-Case reports, letters, conference abstracts, and non-peer-reviewed sources.

The study selection followed a two-phase screening process. Two independent reviewers screened all identified studies based on titles and abstracts. Studies that did not meet the inclusion criteria were excluded. Articles deemed relevant underwent a full-text assessment for eligibility. Any disagreements between reviewers were resolved through consensus or a third reviewer.

Below is the PRISMA flow diagram illustrating the study selection process (Figure 2).

## 3. Results

### 3.1. Vitamin D Synthesis in the Skin

The epidermis is composed of four strata of maturing keratinocytes. Stem cells in the stratum basale (SB), upon the basal lamina, divide the dermis and epidermis. Cells in the top differentiation layers are in contact with growing cells, the spinous layers (SS) overlies basal cells, and keratins K1 and K10 are synthesized at this stage [14]. Profilaggrin and loricrin are two important forms of electron-dense keratohyalin granules in the SS. Profilaggrin helps keratin filaments aggregate. Following the SG, organelles disintegrate, and a water-insoluble, resistant substance forms around keratin from filaggrin complexes and nearby extracellular lipids [15]. With inner and outer layers, the outer epidermis prevents infections from entering. The barrier also suppresses the innate immune system, and when it is broken, defensins like cathelicidin kill organisms quickly [16]. In his research, Holick wrote almost forty years ago that UVB photons penetrated the skin and converted 7-dehydrocholesterol, a precursor in the stratum basale and stratum spinosum, to pre-vitamin D3 [17].

The remaining portion of vitamin D3 production (from skin to cholecalciferol) is affected through covalent bonds, which are then broken by thermal energy [18,19]. During the metabolism of vitamin D3, the synthesized molecule goes into the bloodstream, forming a complex with its binding protein [20,21]. It is subjected first to hydroxylation in the liver to yield the primary circulating form 25-hydroxyvitamin D. Later, within the kidney, it acquires still another hydroxyl group to produce the hormonal active form 1,25-dihydroxyvitamin D. While it has many actions in vivo, the key role among its demonstrated range of physiological processes is in calcium and phosphate homeostasis [22,23].

Figure 3 explains the factors influencing Vitamin D production. Sunscreen use and greater melanin concentrations in the skin reduce UVB radiation to the point where longer times need to be spent in the sun to produce adequate vitamin D [7]. The process of skin changes due to aging makes the skin less efficient in producing vitamin D because aging causes a reduction in the 7-dehydrocholesterol levels. In obese individuals, the bioavailability of vitamin D decreases since it is taken up within the fat tissue; a further impact is that of season and latitude, since UVB radiation is weaker during wintertime and in regions close to the poles [12]. Wearing complete clothing during wintertime and at a given time of day wholly reflects protection measures during the maximum UVB radiation period at noon. At the same time, human-made pollutants screen out UVB rays. and low sun exposure is common in chronic disease and prolonged hospital stays [11]. This all further adds to the deficiency, as shown in Figure 3 by the wide range of interactions on vitamin D synthesis.

#### 3.1.1. Age

MacLaughlin and Holick studied pre-vitamin D3 synthesis in the skin of 8- to 18-year-old and 77- to 82-year-old subjects [24]. The results showed that old age reduces this process more than twofold. This is further supported by evidence in the literature, which states that the concentration of 7-dehydrocholesterol in the epidermis decreases with age: in fact, 70-year-olds have about 50% less vitamin D3 generated in their skin than their 20-year-old counterparts [25,26,27]. This reduction emphasizes the importance of dietary supplementation in older populations to maintain adequate vitamin D levels.

#### 3.1.2. Skin Pigmentation

Despite its cultural and cosmetic relevance, melanin’s significance in skin pigmentation is debatable. Due to the rising incidence of UV-induced skin cancer and ozone layer depletion, which contradicts the public perception of a tan as healthy, a better understanding of melanin’s role in preventing DNA damage and the malignant transformation of skin cells is needed [28]. The higher melanin content in the skin of darkly pigmented persons lowers the efficiency of UVB-induced conversion of 7-dehydrocholesterol to pre-vitamin D3. Hence, darkly pigmented individuals often require more UVB exposure time to produce the same amount of vitamin D as lighter-skinned individuals [29,30].

#### 3.1.3. Geographic Location and Season

Latitude is the most powerful factor concerning the variation in UVB [31]. Thus, in the study of Kallioğlu et al., it was argued that, in the winter season especially, the sun’s angle does not give enough UVB in high-latitude regions to reach the Earth’s surface for the skin production of vitamin D [32]. For equatorial regions, UVB rays consistently reach the Earth’s surface throughout the year; synthesis in these regions is, therefore, perennial [18,33].

#### 3.1.4. Sunscreen Use and Clothing

The use of high-sun-factor sunscreens drastically decreases the UVB that reaches the skin and, therefore, the cutaneous synthesis of vitamin D. High-factor clothing has similar activity. Both can be used as protective means in skin cancer, but they do not allow UV radiation, which is instead required because cutaneous cells can hardly synthesize enough vitamin D for the body’s needs from dietary or supplementary vitamin D [34].

#### 3.1.5. Environmental Factors

The variables that act on UVB are pollution, cloud cover, and heavy cloud clover that suppress UVB to a lesser intensity. In contrast, a thinner atmosphere provide much easier access to penetrating UVB, which is much higher in energy [35].

### 3.2. Something New Under the Sun

Revolutionary research [36] by Andrzej et al. proposes a new hypothesis that challenges the widely accepted perspective of vitamin D3’s biological effects. They propose that 1,25(OH)2D3 may extend these effects beyond traditional VDR. In contrast to the “VDR-centric” perspective, a look at alternate vitamin D routes and novel secosteroidal derivatives signaling via other nuclear receptors suggests a much broader multi-faceted role for vitamin D. This unique technique allows for vitamin D metabolite therapy research.

The authors [36] dispute whether all “nonclassical” biological effects of vitamin D3 are mediated by binding its active metabolite, 1,25-dihydroxyvitamin D3, to the VDR. This approach, which dominates the literature, is under attack from rising evidence. These studies demonstrate that secosteroidal derivatives of D3 and other routes can have substantial biological effects without VDR involvement.

The schematic overview in [36] highlights D3 metabolism’s intricacy, revealing various routes to bioactive metabolites. UVB exposure converts 7-dehydrocholesterol into pre-vitamin D3, which thermally isomerizes into D3. Traditionally, only the first canonical pathway has been examined for biological effects.

### 3.3. Biochemical Pathways of Lumisterol and Tachysterol

After prolonged exposure to solar radiation, 7-DHC converts to biologically inactive lumisterol and tachysterol. High concentrations of UVB photochemically change 7-dehydrocholesterol into lumisterol (L3), a stereoisomer [11]. By enzymatic hydroxylation, CYP11A1 produces 20(OH)L3, 22(OH)L3, 20,22(OH)2L3, and 24(OH)L3. Hydroxylumisterols, as reverse agonists of RORα/γ, interact with the vitamin D receptor’s non-genomic binding site. Intracellular receptors mediate photoprotection and anti-inflammatory action [17].

Tachysterol (T3), another UVB-induced 7-DHC photoproduct, is in dynamic equilibrium with pre-vitamin D3. It may bind to vitamin D-related pathways due to its greater flexibility compared to vitamin D3 and lumisterol.

Lumisterol forms hydroxy-lumisterols like 20(OH)-L3 and 22(OH)-L3. These metabolites regulate skin homeostasis and immunological responses [21]. Additional enzymatic reactions produce lumisterol-derived secosteroids, which resemble vitamin D analogs.

CYP27A1, CYP11A1, and CYP24A1 hydroxylate tachysterol to 20(OH)-T3. Hydroxylated tachysterols may operate as vitamin D metabolites and interact with VDR and RARs [30].

In a study conducted by Schummer et al. [37], after 15 min of skin exposure, pre-vitamin D3 was the primary photoproduct, reaching 15% of the skin’s pre-irradiation 7-dehydrocholesterol content after 30 min. After 1 h, tachysterol reached 5% of the initial 7-dehydrocholesterol concentration due to longer exposure times. After 8 h of constant irradiation, skin lumisterol concentrations reached 50% of 7-dehydrocholesterol levels. After 1 h of exposure, the amount of lumisterol produced from ergosterol reached 50% of ergocalciferol and 25% of tachysterol.

Figure 4 provides a comprehensive overview of the enzymatic pathways and synthesis sites involved in vitamin D metabolism, highlighting the role of cytochrome P450 enzymes in generating various bioactive vitamin D metabolites. It outlines the transformation processes from precursor molecules (7-dehydrocholesterol, pre-vitamin D3, and D3) to hydroxylated derivatives, detailing their enzymatic modifications and locations of synthesis, such as the skin, liver, kidney, and adrenal cortex.

Figure 4 describes the complex metabolic pathways that synthesize and change vitamin D metabolites and their derivatives, including enzymes and synthesis sites. Under UVB exposure, skin 7-dehydrocholesterol (7DHC) becomes pre-vitamin D3 (pre-D3). CYP11A1 (made in adrenal cortex mitochondria) and CYP27A1 (produced in liver mitochondria) catalyze further hydroxylation, converting pre-D3 to T3 and other hydroxylated metabolites. CYP24A1 in the kidneys enables the multistep hydroxylation of D3, resulting in advanced forms like 20,23(OH)_2_D3 and 20,25(OH)_2_D3. Furthermore, endoplasmic reticulum enzymes in the ES route metabolize 7DHP and 20(OH)7DHC to produce 17,20(OH)_2_,7DHP. These complex mechanisms, including tissue-specific enzyme synthesis, demonstrate how the skin, liver, kidney, and adrenal cortex regulate vitamin D metabolism and its physiologically active derivatives. This detailed dissection helps explain the enzymatic details needed to maintain physiological equilibrium.

Vitamin D metabolites play other roles through noncanonical mechanisms. These pathways start with CYP11A1 hydroxylating D3, L3, and 7DHP. The derivatives, including 20(OH)D3 and 22(OH)D3, as well as their further hydroxylated forms, are active compounds that act independently of the VDR. Instead, these metabolites seem to cross-react with other nuclear receptors, including retinoid-related orphan receptors RORα and RORγ, aryl hydrocarbon receptor AhR, and liver X receptor LXR, among others, to modulate an abundance of signaling pathways.

For instance, these derivatives block the NF-κB signaling pathway and also modulate hedgehog plus wnt/β-catenin pathways, which are key in cellular differentiation and immune response to cancer development. In addition, VDR-independent actions, such as antivirus effects and control of oxidative stress via NRF2 regulation and control of DNA repair via p53 phosphorylation, also contribute to the effects.

The authors also show how noncanonical pathways differ from the established ones in the generation of metabolites [36]. Thus, while in canonical metabolism 1,25(OH)2D3 is the center of focus, noncanonical pathways yield a much wider scope of metabolites, many notably originating from lumisterol and tachysterol. These substances, previously ignored as having no biological significance, are have now been discovered to be potent prohormones with useful therapeutic applications.

This evidence puts into question the widely held view that 1,25(OH)2D3 is the only metabolite of D3 that has biological relevance. Instead, it indicates that VDR activation is only one of the functional series of D3. The discovery of many metabolites and their capacity to engage with different receptors should broaden our understanding of the nature and functions of D3 in health and diseases. Such knowledge grants the possibility of targeting new therapeutic techniques, especially because noncalcemic and nontoxic derivatives may be useful in the treatment of inflammation, cancer, and autoimmune diseases.

## 4. Discussions

### 4.1. The Skin as a Metabolic Organ for Vitamin D

Barrier skin is gradually being viewed as the primary organ of systemic and local regulation of vitamin D. Vitamin D is synthesized, metabolized, and directed within keratinocytes, the major cells of the epidermis [31].

#### 4.1.1. Keratinocytes and Vitamin D Metabolism

As indicated, mitochondria and vitamin D are linked. Primary vitamin D3 metabolism is carried out by inner mitochondrial membrane heme-containing cytochromes CYP27A1, CYP27B1, CYP24A1, and CYP11A1 [38]. 

Figure 5 shows that besides the nucleus and cytoplasm, human platelets, megakaryocytic cells, keratinocytes, and fibroblasts have mitochondria steroid receptors and VDR [23]. Though proliferative cancer cell lines have various forms of pro-estrus transforming growth factor receptor localization, mitochondrial VDR is ligand-independent. Due to the lack of an N fragment mitochondrial targeting sequence, VDR may be transported via PTP instead of TOM/TIM translocase [39,40].

The steroid receptor coactivator (SRC) is helpful in the differentiation of keratinocytes and hair, proliferation inhibition, and cancer protection, while the vitamin D receptor-interacting protein (DRIP) facilitates barrier acquisition and antimicrobial defense [41]. It is regulated through very complex genes that are major actors in the process of keratinocyte differentiation implicated in the regulation of gene expression for the control of genetic expression. ABCA12, UGCG, and ELOVL4 are implicated in the regulation of the cell cycle and epidermal structure of cyclin D1, G1, K1, K10, FLG, and LOR [42]. Other genes related to further cancer protection are P53 and GADD45. While CYP27B1 is still controlled in keratinocytes, it is stimulated by calcium, PTH, interferon-gamma (IFN-gamma), and tumor necrosis factor alpha [43]. Upon its synthesis, 1,25(OH)2D binds to VDR located within the nucleus of keratinocytes and epidermal cells, thus functioning as a hormone [44].

#### 4.1.2. Local Production and Role of Vitamin D in the Skin

Besides systemic endocrine actions, the vitamin D produced locally in the skin executes paracrine and autocrine actions [45]. The autocrine action of 1,25(OH)2D includes a direct response on keratinocytes, wherein, among other actions, it has a role in cellular proliferation, differentiation, and apoptosis [46]. Vitamin D increases stratum corneum turnover by differentiating corneocytes from keratinocytes. It regulates keratinocyte growth, which is downregulated after a specific level to balance corneocyte accumulation and shedding, required for epidermal turnover and repair [47].

Skin-resident immune cells are affected by vitamin D’s paracrine actions. Keratinocytes generate 1,25(OH)2D, which regulates Langerhans and cutaneous dendritic cells [48]. In psoriasis, vitamin D analogs control hyperproliferation and immunological dysregulation [23].

#### 4.1.3. Immunomodulatory and Barrier Functions of Vitamin D in the Skin

After microorganisms invade the immune system, 1,25(OH)2D regulates antimicrobial peptide (AMP) expression to produce them. AMPs like cathelicidin and β-defensin boost keratinocyte-mediated immunity. Along with cathelicidin and β-defensin, they boost immunity [49,50,51]. They modulate inflammatory cytokines and inhibit IL-17 and IFN-γ production, contributing to skin disease etiology [52,53]. This is addressed by atopic dermatitis and other dermatosis treatments [54].

Meanwhile, 1,25(OH)2D induces keratinocyte development, which forms an epidermal barrier that inhibits water loss and environmental aggressiveness [55]. The intercellular cohesion needed for this is achieved through the expression of tight-junction proteins modulated by vitamin D [56]. Moreover, in the context of wound healing, 1,25(OH)2D promotes the migration and proliferation of keratinocytes and fibroblasts, facilitating re-epithelialization [57].

Any abnormality in vitamin D metabolism severely impacts skin health. Its insufficiency increases the risk of psoriasis, atopic dermatitis, and other chronic wounds [58]. This compound controls abnormal keratinocyte proliferation and differentiation in psoriasis [59]. Topical calcipotriol accentuates keratinocyte-mediated vitamin D metabolism [60]. In contrast, excessive pathway activation can produce hypercalcemia or paradoxical epidermal differentiation [61].

### 4.2. The Dual Role of the Skin in Vitamin D Physiology

Vitamin D is produced mostly in the skin and has biological effects. This dual role sustains keratinocyte activity, wound healing, and barrier integrity.

#### 4.2.1. Skin as a Producer of Vitamin D

Considered a unique endocrine organ, skin manufactures vitamin D. While the diet contributes a small percentage, sunshine creates 80–90% of the vitamin D found in the skin. The skin controls vitamin D synthesis [61]. After adequate synthesis, pre-vitamin D₃ photodegrades into lumisterol and tachysterol, lowering sun-induced vitamin D toxicity [62]. This self-regulating mechanism makes cutaneous vitamin D synthesis safe and effective [63]. Furthermore, some findings suggest a role of vitamin D skin synthesis in the microbiome. In this way, the absorption of UVB might be regulated by the bacteria existing in the skin, as well as some local metabolic pathways [64].

#### 4.2.2. Skin as a Target for Vitamin D

Among others, the key local actions of vitamin D on the skin are the following: It regulates keratinocyte behavior, balances the immune response, supports barrier functions, including the expression of defense molecules, initiates re-epithelialization in wound repair, and provides protection against carcinogenesis [40]. The loss of any function results in homeostatic imbalances, thus causing cutaneous pathologies (i.e., psoriasis and hyperproliferative keratinocytes) [44]. Cathelicidin is an AMP with healing purposes. The skin’s defensive function may also largely depend on the proper functioning of the stratum corneum and tight junctions among keratinocytes [65,66]. Indeed, impaired vitamin D signaling can compromise this barrier, enhancing susceptibility to infections and dehydration [67].

### 4.3. Systemic Effects of Vitamin D Deficiency and the Skin’s Contribution

Vitamin D deficiency has joined a wide variety of systemic diseases, spanning from autoimmune diseases, osteoporosis, and cardiac diseases to neuropsychiatric diseases and metabolic syndromes [68,69,70]. Vitamin D’s dual role as the primary site for synthesis and a target of its multiple biological actions creates a sophisticated feedback loop between systemic levels of vitamin D and cutaneous synthesis.

#### 4.3.1. Immunological Impacts: Autoimmune Diseases

An active form of vitamin D can regulate immune functions. Indeed, it acts immediately upon binding to the special receptor of VDR expressed by several immune cells, such as T lymphocytes, B lymphocytes, macrophages, and dendritic cells, through which this interaction specifically regulates the innate and adaptive forms of the immune response. Thus, vitamin D is important in maintaining immune homeostasis [71].

These pro-inflammatory cells are often implicated in autoimmune diseases such as rheumatoid arthritis (RA) and systemic lupus erythematosus (SLE) [72].

Emerging evidence indicates the major involvement of vitamin D deficiency in the development and progress of the above conditions, which frequently seem to be in an inverse relationship with the serum levels of 25(OH)D [64,68]. The pathology of most diseases has further identified vitamin D as an active anti-inflammatory agent, specifically by downregulating the two key demyelinating and neuronal destruction cytokines IL-17 and IFN-γ [73]. In systemic lupus erythematosus, reduced vitamin D levels are linked to disease activity, reflecting the loss of immunoregulatory control. Moreover, in rheumatoid arthritis, elevated levels by synovial macrophages and T-cells cause joint inflammation and erosion [48].

#### 4.3.2. Bone Health and Beyond: Implications for Cardiovascular, Metabolic, and Mental Health

As mentioned previously, vitamin D increases calcium absorption, which is essential for bone mineralization and skeleton health [74]. This vitamin deficit causes adult osteomalacia and osteoporosis [75,76]. Hypertension and cardiac dysfunction are linked to vitamin D deficiency [77]. The Journal of the Endocrine Society found that vitamin D and calcium supplementation lower hypertension in obese older adults, benefiting their cardiovascular system. Insulin resistance and type 2 diabetes indicate vitamin D deficiency [78,79]. Vitamin D can produce metabolic syndrome by affecting pancreatic β-cell activity and insulin sensitivity, leading to mood disorders such depression [80]. VDRs in mood-regulating brain areas demonstrate vitamin D’s antidepressant and intracellular signal transduction roles in neuropsychiatric health [81].

### 4.4. The Skin in Vitamin D Supplementation

#### 4.4.1. Oral Supplementation

Oral vitamin D supplementation is the most common treatment for deficiency [82]. It requires administering ergocalciferol or cholecalciferol (capsules, pills, liquids) [83]. A number of other stories confirm that this strategy enhances the serum concentration of 25(OH)D [84,85,86]. In the study conducted by Todd et al., oral vitamin D₃ capsules and 25(OH)D-boosting spray solutions were used and found to be equally effective over a span of 12 weeks in a randomized controlled experiment [87]. However, oral supplementation has its challenges for people with malabsorption syndrome and GI conditions, as these features do not allow vitamin D to be absorbed via oral intake [88]. In such circumstances, different means of delivery could be investigated.

#### 4.4.2. Topical Supplementation

This involves the use of vitamin D gels, creams, and patches to stimulate or aid growth in people through patch-based stimulation, where vitamin D is supplied through the skin [89]. Therefore, this technique is ideal for patients with genuine eroding or absorption difficulties as it does not involve using the gastrointestinal tract [90]. Increasing the serum 25(OH)D concentration more than 30 ng/mL after four months by applying aloe vera gel containing 5000 IU of vitamin D₃ daily has been shown in a trial to be safe and efficient [91]. Nonetheless, there is still ongoing controversy regarding the appropriate use of vitamin D and its form, although quite a few studies support its topical use. The vitamin D molecules could potentially be blocked from penetrating the skin by the skin barrier function, and there is a lack of uniform formulations; this also affects the degree of reliability of this method [92]. Some, like Murphy et al., on the other hand, are less convinced about vitamin patches and maintain that there is very rudimentary proof that skin can effectively transport vitamin D and other vitamins through patches [93].

#### 4.4.3. Ultraviolet Therapy

Vasodilation is induced, and blood flow is recorded following narrowband UVB (NB-UVB) therapy intake. Ultraviolet light in the form of NB-UVB phototherapy induces the dermal tissues of the skin to produce vitamin D [94]. This approach is especially beneficial for individuals with limited sun exposure or conditions like malabsorption syndromes. Research indicates that NB-UVB therapy effectively increases serum 25(OH)D levels. For example, a study found that NB-UVB therapy raised serum 25(OH)D concentrations comparably to high-dose oral cholecalciferol supplementation [95]. However, UV therapy requires some controlled clinical setting to minimize the risk of skin damage and potential carcinogenesis associated with UV exposure. Therefore, it is typically reserved for individuals who cannot achieve adequate vitamin D levels through oral supplementation or dietary intake [96].

The supplementation method chosen should consider factors such as efficacy, safety, compliance, and the overall health condition of the individual receiving supplementation [97,98]. The oral route is generally safe, cost-effective, and convenient; hence, it would work for most people. Topical supplementation would be an option for people with gastrointestinal problems; however, evidence for its efficacy is scant, and therefore, more validation studies need to be carried out. Although UV therapy can work in some specific populations, it is fraught with the usual risks of UV exposure and is a treatment that requires medical supervision.

### 4.5. Clinical Validation vs. Theoretical Frameworks of Vitamin D’s Effects

While numerous preclinical and in vitro studies suggest that vitamin D influences immune function, cardiovascular health, neuroprotection, and cancer biology, the translation of these findings into clinical practice remains variable. Some effects, such as its role in bone health and calcium–phosphorus regulation, are well established through randomized controlled trials and clinical guidelines. However, other proposed mechanisms, including its impact on autoimmune diseases, cardiovascular function, and cancer prevention, remain largely theoretical or supported by observational studies rather than definitive interventional trials. Emerging evidence from human studies has demonstrated associations between vitamin D levels and disease outcomes, but causality remains uncertain, often confounded by factors such as genetic variability, lifestyle influences, and study design limitations.

## 5. Limitations

Despite the comprehensive scope of this review, several limitations must be acknowledged. First, while we have provided an extensive overview of vitamin D metabolism, its biochemical pathways, and physiological roles, the complexity of these mechanisms means that emerging discoveries may refine or challenge current understandings. The field of vitamin D research is continuously evolving, and novel pathways or interactions may not yet be fully elucidated.

Second, this review primarily relies on data from the published literature, which may introduce publication bias, as studies with negative or inconclusive findings are less likely to be reported.

Another important limitation is the variability in vitamin D measurement methodologies across different studies. The assessment of vitamin D levels, particularly 25-hydroxyvitamin D [25(OH)D], can vary depending on the assay technique used, potentially leading to inconsistencies in the reported outcomes. Finally, while we have explored the interplay between vitamin D and various physiological systems, clinical applications remain an area requiring further investigation. Many studies focus on associations rather than causation, and large-scale randomized controlled trials are needed to establish definitive therapeutic recommendations regarding vitamin D supplementation, topical applications, and UVB therapy.

## 6. Conclusions

The skin plays a central role in vitamin D physiology, functioning both as a producer of this essential secosteroid and as a target for its systemic and local actions. The synthesis of vitamin D initiates in the skin with UVB radiation converting 7-dehydrocholesterol to pre-vitamin D₃, setting off a cascade which ultimately yields the biologically active form of vitamin D, 1,25-dihydroxy vitamin D. This may be affected by several factors such as age, pigmentation, geographical location, and lifestyle behaviors like the use of sunscreens, which actually can lead to minimal vitamin D synthesis and hence systemic levels of vitamin D.

Calcitriol modulates the balance of calcium and phosphorus, bone health, immunomodulation, cardiovascular defense, and metabolism. The fact that the form of vitamin D synthesized in the skin impacts the development of keratinocytes and the integrity of barriers and aids in wound healing underscores its critical importance in skin health. The skin’s unique significance in systemic endocrine processes and localized paracrine and autocrine actions is highlighted. Vitamin D metabolism by the skin is used in oral and topical supplementation and UVB phototherapy. Oral supplementation is the most common and successful way, but topical and UVB therapy can help people with malabsorption problems or restricted sun exposure.

Understanding vitamin D physiology has improved, yet topical formulation optimization and UV therapy safety remain unsolved. Further studies are needed to improve these methods and produce vitamin D metabolism guidelines that account for individual heterogeneity.

## Figures and Tables

**Figure 1 nutrients-17-00803-f001:**
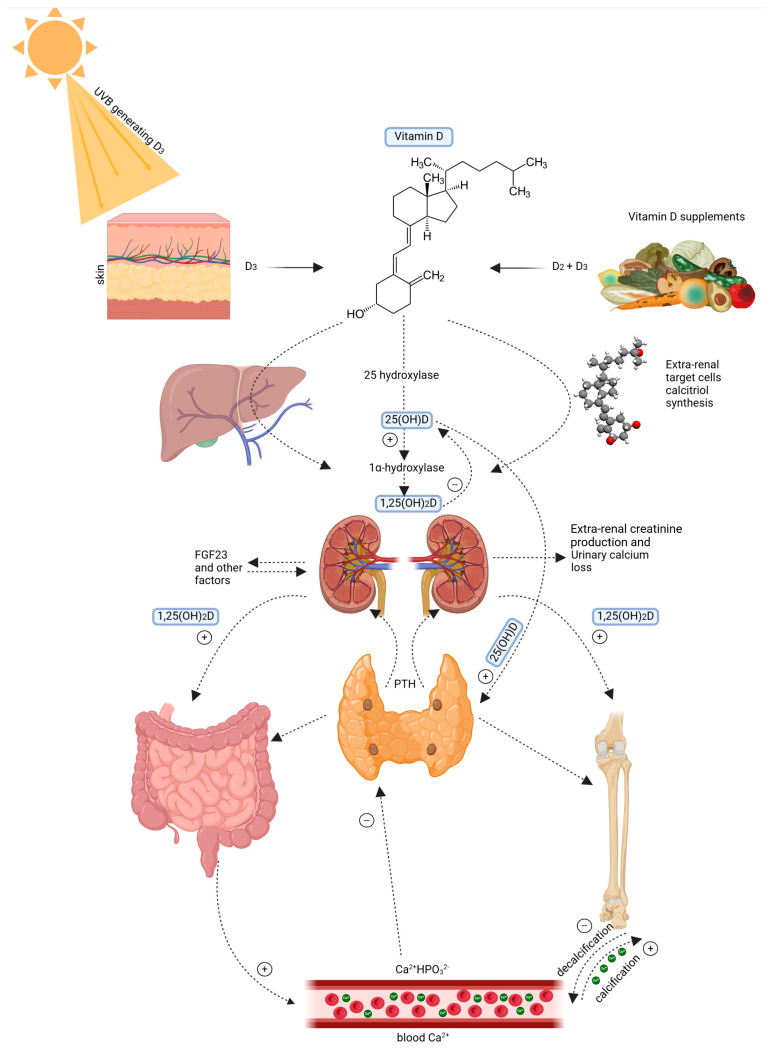
Vitamin D metabolism and systemic regulation pathway. Left: 7-dehydrocholesterol is converted to vitamin D3 due to UVB radiation. Right: Vitamin D is sourced from food in the form of D2 and D3. Vitamin D3 is hydroxylated in the liver into 25-hydroxyvitamin D3 (25(OH)D) and, subsequently, in the kidney into 1α,25 dihydroxyvitamin D3 (1,25(OH)2D). It maintains calcium and phosphate balance by stimulating intestinal uptake, bone building, and altering parathyroid hormone (PTH) release. Feedback mechanisms include fibroblast growth factor 23 (FGF23), urinary calcium excretion, and extra-renal creatinine production. Created with Biorender [10].

**Figure 2 nutrients-17-00803-f002:**
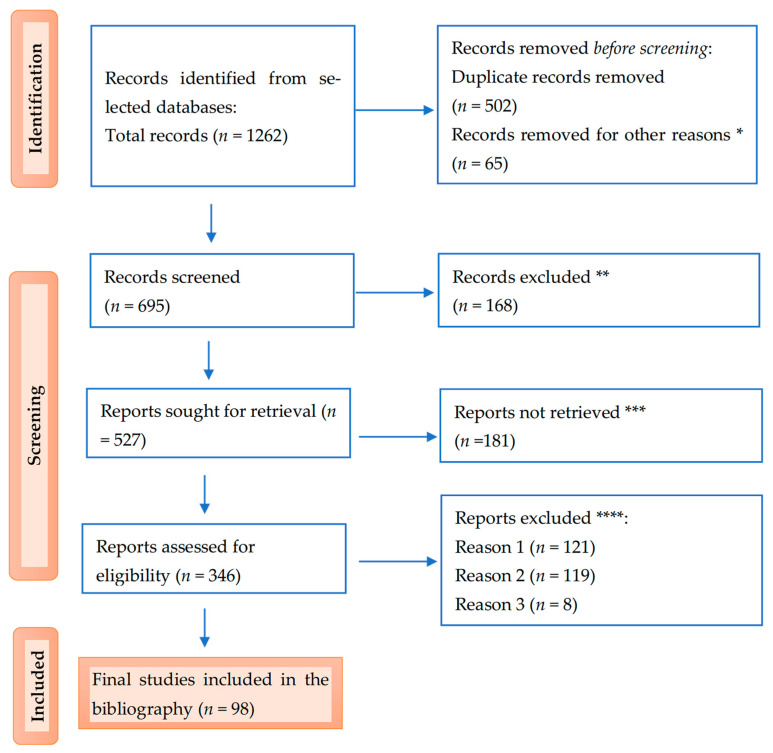
PRISMA flow diagram for the results: * studies are not relevant for the present review; ** studies do not help us provide an answer to the research question; *** unable to find the full text of the study; and **** Reason 1—wrong setting, Reason 2—wrong patient population, and Reason 3—research question not relevant.

**Figure 3 nutrients-17-00803-f003:**
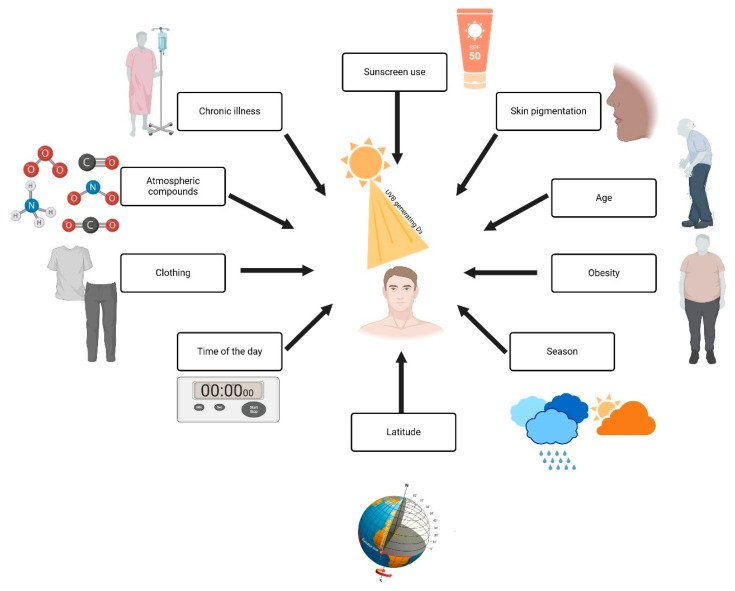
Factors influencing vitamin D synthesis and UVB exposure. Created with Biorender [10].

**Figure 4 nutrients-17-00803-f004:**
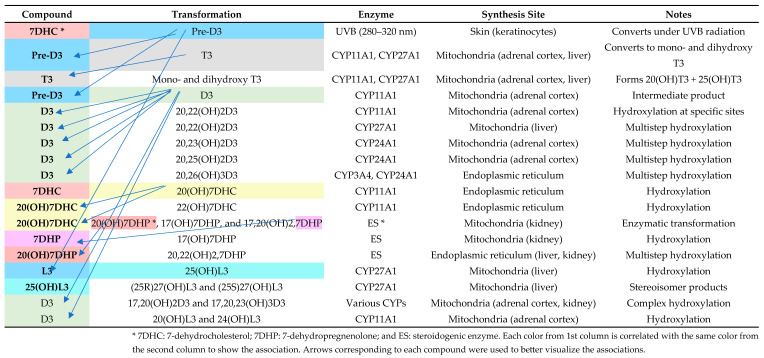
Enzymatic pathways and locations of vitamin D metabolite synthesis. Adapted after Slominski et al. [36].

**Figure 5 nutrients-17-00803-f005:**
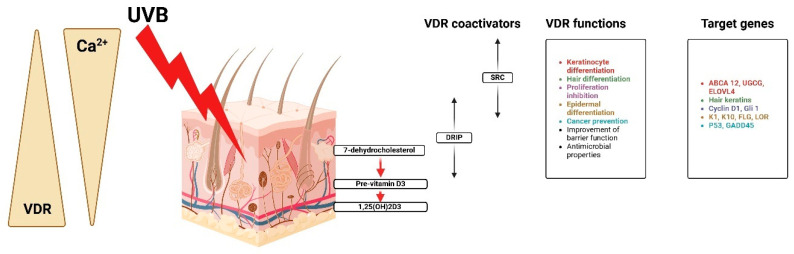
Calcium, VDR, and coactivators regulate keratinocyte activities, proliferation, and differentiation. Created with Biorender [10].

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
