# Peer review of "Vitamin D: Beyond Traditional Roles—Insights into Its Biochemical Pathways and Physiological Impacts"

_nutrients, 2025, doi:10.3390/nu17050803_

Round 1
Reviewer 1 Report
Comments and Suggestions for Authors
This paper focuses the importance of lumisterol and tachysterol in addition to 1,25(OH)2D3. However, this is insufficient in the current form. It would be improved by adding a more detailed introduction about lumisterol and tachysterol and discussing the relevant insights.
1) Please describe the detail of “lumisterol and tachysterol” in the chapter 3 “Something new under the Sun” or in the previous new chapter.
Please provide their chemical structures and “Biochemical Pathways” as indicated in the title of this paper. This will hopefully solve the following problems:
a) No explanation for L3 and T3.
b) What metabolites are lumisterol and tachysterol?
c) L3 and T3 are shown before “lumisterol and tachysterol”(in line 188).
2) Please add information regarding their concentrations in physiological conditions and those required to exert their effects.
3) Please check following sentences.
a) Line 54: “D3 secretion is hydroxylated into 25(OH)D.”
b) Line 57: “extra-renal creatinine production.” Is it included in the “Feedback mechanisms”?
c) Line 141: Can “Andrzej T. Slominski, Robert C. Tuckey, Anton M. 141 Jetten, and Michael F. Holick” be “Andrzej et al.”?
d) Table 1: Caption is missing.
e) Table 1: Please spell out or note “ES”, “DHC”, and “DHP.”
f) Lines 214-215: Whar are “SRC” and “DRIP”?
g) Line 222: “tumor necrosis alpha” should be “tumor necrosis factor alpha.”
h) Lines 238 and 274: “AMP protein” is not common name.
Reviewer 2 Report
Comments and Suggestions for Authors
This is a review paper on known and novel pathways of vitamin D activity. The title and abstract are lively in tone, but describe the basic intent of the paper well.
The introduction gives enough background information on the subject.
However, the structure of the entire paper needs to adhere to PRISMA guidelines. A section describing the literature search used for assembling the paper is mandory, as well as a PRISMA checklist and flowchart.
A limitations paragraph is necessary, as well as completely rewriting the overly long introduction and adhering to the IMRAD organization, since the paper currently reads as a proposed book chapter. I don't mind the lively tone, but the organization of information must be executed well, and a labeling of data related evidence must be added. A PICOS/PICOT analysis table is required and Oxford Level of Evidence Guidelines must be referenced.
Reviewer 3 Report
Comments and Suggestions for Authors
Very interesting and important article I recommend it for publication after referring to my comments.
Areas for Improvement and Recommendations
1. Clarity and Conciseness
Issue: Some sections contain long, complex sentences that can reduce clarity.
Example:
“Aren’t the actions of cytochrome P450 CYP enzymes, alongside lumisterol and tachysterol, new forms of treatment for inflammation, immune system disorders, and cancer?”
Recommendation: Rewrite rhetorical questions as declarative sentences to maintain academic tone.
Suggested revision:
“Recent studies suggest that cytochrome P450 CYP enzymes, lumisterol, and tachysterol may offer novel therapeutic approaches for inflammation, immune system disorders, and cancer.”
Issue: The introduction could be more concise, as it reiterates some points made later in the article.
Recommendation: Summarize the key contributions of vitamin D beyond skeletal health in fewer words and move in-depth discussions to later sections.
2. Strengthening Critical Analysis
Issue: The discussion on non-canonical pathways is promising but lacks critical evaluation of their clinical significance. While the article presents alternative pathways involving RORα, RORγ, and AhR, it does not discuss:
The current limitations in research.
Whether these findings have been tested in human models or remain mostly theoretical.
Recommendation:
Add a brief paragraph addressing:
The need for further research to validate these alternative pathways.
Any conflicting evidence or limitations in current studies.
3. Consistency in Terminology
Issue: The article alternates between “vitamin D,” “D3,” and “cholecalciferol” inconsistently.
Recommendation: Use consistent terminology, specifying when referring to the general concept of vitamin D versus its specific metabolites.
4. Figures and Table Formatting
Issue: Some tables (e.g., Table 1 on enzymatic pathways) contain a wealth of information but are not explained in the main text adequately.
Recommendation:
Before introducing a table, briefly describe its significance.
Use bullet points or separate paragraphs to make interpretations easier.
5. Expanding the Conclusion
Issue: The conclusion effectively summarizes the key points but lacks forward-looking insights on clinical implications.
Recommendation:
Mention specific areas for future research, such as clinical trials testing the efficacy of vitamin D derivatives.
Briefly discuss potential applications in personalized medicine for vitamin D supplementation strategies.
Overall Verdict
Strengths: Well-researched, structured logically, includes recent literature and useful figures.
Areas to Improve: Conciseness, clarity, critical discussion on non-canonical pathways, terminology consistency, and enhancing conclusions.
Final Recommendation: With minor refinements, this article could be a strong contribution to current literature on vitamin D's expanding roles.
The topic was important during COVID19, so there is an important article to refer to because people with normal vitamin D levels were less likely to get sick or the signs and symptoms were less severe.
Associations of the COVID-19 burden and various comorbidities of different ethnic groups in Israel: a cross-sectional study. Clinical and Experimental Medicine.
Round 2
Reviewer 2 Report
Comments and Suggestions for Authors
The authors have undertaken the necessary edits and improved the paper enough for publication. I would congratulate the authors on their work and support publication.